# Land Use Change in the Major Bays Along the Coast of the South China Sea in Southeast Asia from 1988 to 2018

**Junjue Zhang [1,2] and Fenzhen Su [1,2,*]**

[1]  State Key Laboratory of Resources and Environmental Information System, Institute of Geographic Sciences and Natural Resources Research, CAS, Beijing 100101, China; zhangjj@lreis.ac.cn

[2]  Collaborative Innovation Center for the South China Sea Studies, Nanjing 210023, China

*  Correspondence: sufz@lreis.ac.cn; Tel.: +86-010-64888956

**Abstract:** Bays are some of the core areas for marine economic development. The South China Sea coast is one of the most developed and dynamic places in the Asia-Pacific. In this study, we focused on the large bays surrounding the South China Sea. The techniques of image segmentation and supervised classification as well as image interpretation were used to acquire land-use data of 41 bays from 1988 to 2018. Then, we quantified the intensity and pattern of land-use and land-cover change during the two periods. Plantation land was the dominant agriculture land type as well as the second land use type after natural forest. Agriculture land cover increased from 29.8% to 40.9% and the growth was driven by plantation expansion. Deforestation was serious, including both natural forests and mangroves. Natural forest cover decreased by 31.6% and mangrove cover decreased by 16.2%. The vast majority of forest loss occurred in Sumatra and western Kalimantan. Commodity-driven deforestation for plantations was the major reason for forest loss.

**Keywords:** bay; land-use and land-cover change; South China Sea; forest

## 1. Introduction

Bays are inlets that extend into the mainland or islands. According to the causes, the bay can be divided into: structural bay, estuary bay, lagoon bay, bedrock erosional and accumulational bay, and delta bay, etc. [1]. The bay is a multi-layered complex system with abundant resources and numerous elements, which is located at the interface between the sea and the land [2]. As pearls of the coastal zone, the gulf is the most dynamic and prominent geographical area in the coastal zone, as well as the most vulnerable area. The coast around the South China Sea is located at an intercontinental crossroads, which has developed maritime trade, prosperous tropical economic crops, and rich mineral resources. It is one of the regions with the highest potential for economic development in the Asia-Pacific. Over the past century, the Southeast Asian regions have experienced rapid urbanization and industrialization [3]. Correspondingly, as one of the core areas of economic development, the bays surrounding the South China Sea have been subjected to serious environmental stresses. Problems such as overexploitation of land resources, overdraft of groundwater, untreated municipal wastewater, and habitat destruction, etc., directly affected biodiversity, biogeochemical cycle, sustainable utilization of resources and environment, and human production and living activities. Study of the land-use and land-cover change (LUCC) in this region may provide basic data support for the research of human activities on coastal ecosystem, coastal land resource utilization, and environmental change [4,5].

International organizations have carried out many LUCC studies in Southeast Asia. In 1994, the United Nations Environment Program (UNEP) launched the "land cover assessment and simulation"

(LCAM) project for Southeast Asia to study the current situation and change of land cover in Southeast Asia and identify the hot spots of local land cover change through the study [6]. The global environmental center of Japan National Academy of Sciences proposed the land use research for global environmental protection project (LU/GEC) [7]. The project focuses on sustainable land use in the Asia-Pacific region, including the spatial distribution, temporal dynamics, and drivers of LUCC.

Many scholars around the world have conducted research in this area. The current studies focus on three fields: rapid urbanization, crop booms, and deforestation. Schneider et al. [8] discussed the urban landscape change in East and Southeast Asia between 2000 and 2010. Results showed that urban land increased >22% while urban populations climbed >31%. Studies in Metropolitan Regions such as Guangzhou, Hanoi, Ho Chi Minh, Bangkok, and Manila have shown that such cities were facing continuous and dramatic urban expansion problems [9–12]. Built-up areas in Guangzhou-Foshan increased by 64% from 1990 to 2010, built-up land in Haoni increased by 25% from 1993 to 2010, built-up areas in Ho Chi Minh increased to 4.8 times their initial size from 1990 to 2012, urban/built-up land in Bangkok increased by 186% from 1988 to 2009, and urban/built-up land in Metro Manila increased by approximately 50% from 1993 to 2009 (Table 1). The needs of economic development and the increasing pattern of population migration to coastal areas led to rapid urbanization. It also brought a series of problems including, but not limited to, urban thermal effects, flooding risks, and climate change [13,14].

**Table 1.** Built-up area increases in metropolitan regions surrounding the South China Sea.

| City | Increase Pace | Increased Area | Time Interval |
| --- | --- | --- | --- |
| Guangzhou-Foshan | 64% | 1142 | 1990 to 2010 |
| Haoni | 25% | 128 | 1993 to 2010 |
| Ho Chi Minh | 4.8 times | 650 | 1990 to 2012 |
| Bangkok | 186% | 946 | 1988 to 2009 |
| Metro Manila | 50% | 145 | 1993 to 2009 |

Research of crop booms, primarily those of plantation crops, has become one of the most common topics in the study of Southeast Asia LUCC. Industrial plantation forestry took an important place in the long history of 'boom' natural resource sectors in Southeast Asia [15]. The global planted forest area increased from 167.5 million ha in 1990 to 277.9 million ha in 2015. Over the period, the planted forest area in Southern and Southeast Asia increased from 15.9 million ha to 29.9 million ha, with a rapid growth rate of 85% [16]. Rubber and oil palm are the two main plantation crops in Southeast Asia. Results in the lowlands of Peninsular Malaysia, Borneo, and Sumatra for 2010 showed that the area of closed canopy oil-palm plantations summed up to 8.3 million ha. Among these areas, one-tenth of these plantations were established on peatlands. Human activity in tropical peatlands may affect the global carbon cycle and increase the annual fire risk [17,18]. The FAO (Food and Agriculture Organization of the United Nations) statistics for the period from 2000 to 2010 revealed that rubber was the most rapidly expanding tree crop in the five Montane Mainland Southeast Asia countries (Cambodia, Laos, Myanmar, Thailand, and Vietnam) [19]. Li et al. [20] developed a map of the rubber tree growth in mainland Southeast Asia based on time-series MODIS (Moderate-resolution Imaging Spectroradiometer) 250 m NDVI (Normalized Difference Vegetation Index) products and sub-national statistical data. Compared to the GLC 2000 (2003) land-cover map, most of the new rubber trees were found in northern Laos, eastern Myanmar, and northeastern Thailand.

Southeast Asia is rich in forest resources, and about 15% of the world's tropical forests were located in the region [21]. Additionally, the deforestation rate of Southeast Asia was high in tropical countries. Miettinen et al. [18,22] produced a 250 m spatial resolution land cover map of Southeast Asia for 2010 and 2015. The map provided a detailed forest classification scheme, including mangrove, peat swamp forest, lowland forest, lower montane forest, upper montane forest, and plantation/regrowth. Dong et al. [23] generated the first 50 m forest cover map of Southeast Asia in 2009. The 50 m forest

map was expected to provide more effective support for the evaluation of forest fragmentation. Stibig et al. [24] discussed the change in tropical forest cover of Southeast Asia from 1990 to 2010. The results showed that the forest-covered area of Indonesia decreased from 123.8 million ha in 1990 to 104.4 million ha in 2010, accounting for a large proportion of the total decreased forest. As a globally important carbon pool, peatlands' exploration in Southeast Asia has attracted much attention [25,26]. Miettinen et al. [27] analyzed deforestation rates in insular Southeast Asia between 2000 and 2010, and the results showed that peat swamp forests clearly experienced the highest deforestation rates, with an average annual rate of 2.2%. In addition, Southeast Asia contains more than one-third of the world's mangrove forest. Research on mangrove deforestation showed that mangrove decreased by 2.12% from 2000 to 2012 [28]. Mangrove forests in local areas, such as the Red River Delta, Mekong Delta, Kuching, etc., have been seriously damaged due to aquaculture, plantation expansion, and urbanization [29–31].

According to the results of a literature search, the current land-use studies focus on the large-scale regional research in Southeast Asia or small-scale local research in the coastal delta plain, the Strait of Malacca, and the highly developed urban areas. As the highly developed and vulnerable coastal areas of the South China Sea, gulfs deserve systematic study. In this paper, we aimed to provide a uniform assessment of the land use and changes of the large bays surrounding the South China Sea (not including China) for the period from 1988 to 2018. Forty-one large bays (water area > 50 square kilometers) were selected as typical coastal areas for analysis of the spatial and temporal distributions. In view of the widely discussed issues in land research in Southeast Asia, four types of typical bay features, namely construction land, agricultural land, natural forest, and mangrove forest, were discussed in detail at the region level, country level, and bay level. Additionally, we attempted some comparisons between the bays and the whole of Southeast Asia.

## 2. Materials and Methods

### 2.1. Study Area

The South China Sea is the largest marginal sea in the Northwest Pacific, covering more than 3 million square kilometers of area [32]. It is located in a tropical and semitropical region, roughly between the latitudes of 4° S and 24° N and the longitudes of 102° E and 123° E (Figure 1). Geographically, the South China Sea is bordered by the South China Continent to the north, the Central South Peninsula and the Malay Peninsula to the west, the Sumatra and Kalimantan Islands to the south, and the Philippine Islands to the east. There are nine countries around the South China Sea, including China, Vietnam, Cambodia, Thailand, Malaysia, Singapore, Indonesia, Brunei, and the Philippines.

Topographically, Southeast Asia is divided into two parts: the Indochina Peninsula and the Malay Archipelago. The Indochina Peninsula extends southward to the southern mountain range of China. The terrain is high in the north and low in the south, and the land mass is gradually decreasing. The large alluvial plains are in the lower reaches of the river. The terrain of the Malay Islands (including Malay Peninsula, Sumatra, Kalimantan, and the Philippine Islands) is dominated by mountains with narrow plains along the coast [33].

The Indochina Peninsula and Philippine Islands in the north have tropical monsoon climates. Precipitation has obvious seasonal changes, and the climate can be divided into wet and dry seasons. The southern region has a tropical rainforest climate, abundant rainfall, and perennially high temperature. The annual average temperature is 25–28 °C, and the annual expected rainfall is more than 2000 mL. Good climate conditions make Southeast Asia one of the world's major rice producing regions and the largest tropical economic crop producing region. Indonesia, Thailand, and Vietnam are major rice producers [34]. Thailand is the world's largest rubber producer and exporter [35]. Malaysia and Indonesia are the world's first and second largest oil palm producers and exporters, and the Philippines is the world's largest coconut producer and exporter [36].

There are 41 large bays (water area > 50 square kilometers) around the South China Sea (Table 2), including 14 bays on the Indochina Peninsula, 4 bays on the Malay Peninsula, 8 bays on Sumatra, 7 bays

on Kalimantan, and 10 bays on the Philippine Islands. Among these bays, the largest bay (Kuching Bay, Malaysia) covers a water area of 13,945 square kilometers. Next, Bangkok Bay (Thailand) and Kuala Tungkai Bay (Indonesia) take second and third place respectively, with 9710 square kilometers and 6249 square kilometers of water area. Bays with water areas of between 1000 and 3000 square kilometers are Kompong Som Bay (Cambodia), Songkhla Bay (Thailand), Sukadana Bay (Indonesia), Brunei Bay (Brunei), Manila Bay (Philippines), and Lingayan Bay (Philippines). Many famous port cities are distributed in the bays, including Hai Phong, Sihanoukville, Bangkok, Songkhla, Belawan, Kuching, and Manila.

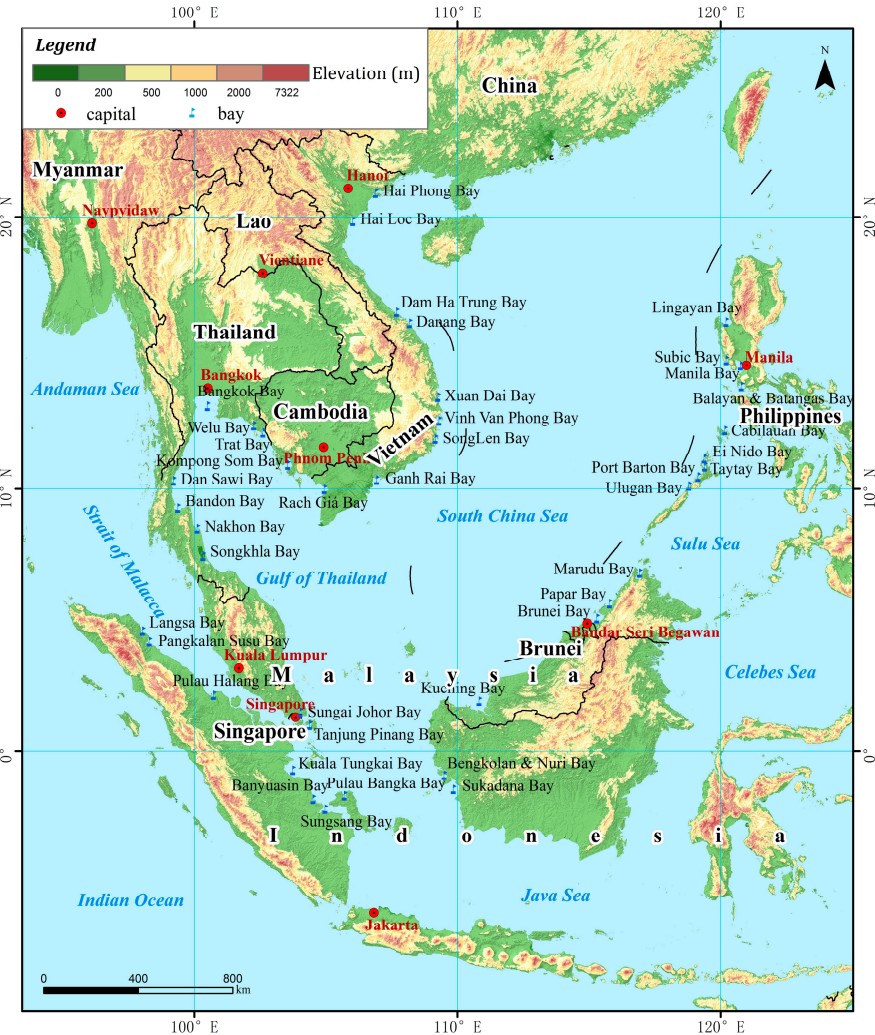

**Figure 1.** Location of bays surrounding the South China Sea.

**Table 2.** Major bays surrounding the South China Sea.

| Country | CT | Name | Location | Country | CT | Name | Location |
|---------|----|------|----------|---------|----|------|----------|
| Vietnam | 9 | Hai Phong Bay | Hai Phong | Philippines | 9 | Ulugan Bay | Puerto Princesa |
| | | Hai Loc Bay | Hai Loc | | | Port Barton Bay | Port Barton |
| | | Dam Ha Trung Bay | Hue | | | Taytay Bay | Taytay |
| | | Danang Bay | Danang | | | Ei Nido Bay | Ei Nido |
| | | Xuan Dai Bay | Song Cau | | | Cabilauan Bay | Cabilauan |
| | | Vinh Van Phong Bay | Ninh Hoa, Nha Trang | | | Balayan & Batangas Bay | Balayan, Batangas |
| | | SongLen Bay | Cam Ranh | | | Manila Bay | Manila |
| | | Ganh Rai Bay | Vung Tau | | | Subic Bay | Olongapo |
| | | Rạch Giá Bay | Rạch Gia | | | Lingayan Bay | Lingayan |
| Cambodia | 1 | Kompong Som Bay | Sihanouk | Brunei | 1 | Brunei Bay | Seri Begawan |
| Thailand | 7 | Trat Bay | Trat | Indonesia | 10 | Langsa Bay | Langsa |
| | | Welu Bay | Ban Khlung | | | Pangkalan Susu Bay | Pangkalan Susu |
| | | Bangkok Bay | Bangkok | | | Pulau Halang Bay | Bagan-siapiapi |
| | | Dan Sawi Bay | Chumphon | | | Kuala Tungkai Bay | Jambi |
| | | Bandon Bay | Surat Thani | | | Sungsang Bay | Sungsang |
| | | Nakhon Bay | Nakhon | | | Banyuasin Bay | Banyuasin |
| | | Songkhla Bay | Songkhla | | | Tanjung Pinang Bay | Tanjung Pinang |
| Malaysia | 4 | Sungai Johor Bay | Johor | | | Pulau Bangka Bay | Belinyu |
| | | Kuching Bay | Kuching | | | Bengkolan & Nuri Bay | Telukbatang |
| | | Papar Bay | Papar | | | Sukadana Bay | Sukadana |
| | | Marudu Bay | Marudu | | | | |

Note: "CT" is the abbreviation of "Count".

### 2.2. Data Sources

The data used in this study included the Landsat Thematic Mapper images captured in 1988 and 2018. The images were downloaded from the US Geological Survey Center for Earth Resources Observation and Sciences (USGS, https://earthexplorer.usgs.gov/) at Level 1. In total, we collected 99 images, including 49 Landsat 5 TM (Thematic Mapper) images and 50 Landsat 8 OLI (Operational Land Imager) images. Standard radiometric and geometric corrections were applied. The images were selected because they were of good quality and the bay areas were free of cloud.

### 2.3. Image Analysis

Before we analyzed the bays, the first question was how to define the scope of the bay. Then, a land use classification system was constructed, and a classification method was introduced.

#### 2.3.1. Bay Scope

When defining the scope of the bay, we used the following strategies (Figure 2):

- The bay area consisted of the land area and sea area.
- The land area was defined as landward area extending 10 km from the coastline.
- The sea area was defined as the waters enclosed by the coastline and the entrance of the bay.
- For the coasts with mangroves, the coastline lies on the landward vegetation line. This delineation can help with the analysis of the complete change in mangrove area in the bays.

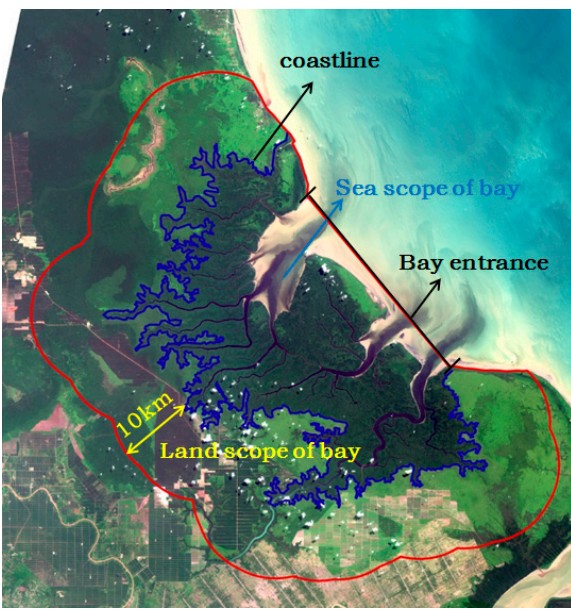

**Figure 2.** Scope of bay.

### 2.3.2. Land Use Classification System

Scientists around the world have carried out studies on classification systems of land use under different resolutions in coastal areas. Compared to the traditional land use classification system, the classification system of coastal land use is more detailed in the coastal wetland. The coastal wetland can be subdivided into natural wetland and artificial wetland [37,38]. Natural wetlands include beaches, marshes, mangroves, estuaries, and lagoons, etc. Artificial wetlands include culture ponds, salt fields, paddy fields, and reservoirs, etc. In addition, there are some typical land use types in coastal zone, such as open beaches, vegetated dunes, and bare rocks, etc. [39,40].

According to the research purposes, research scales, and images adopted in this paper, the following classification system is formulated based on the existing classification system mentioned above as well as the regional characteristics of Southeast Asia. The land use was mapped at two levels. At the primary level, there were eight categories: cultivated land, forestland, grassland, construction land, industrial land, water, coastal wetland, and unused land. To analyze the typical land use types in the gulfs, the coastal wetlands were further divided into mangroves, salt fields, culture ponds, tidal flats, and sea waters. In addition, as a widely discussed research topic in the region, the forestlands were subdivided into natural forests and plantations for further analysis. Together, fifteen land use types were identified at the secondary level and our classification is explained in Table 3.

**Table 3.** Land use categories.

| Level 1 | Level 2 | Abbreviation | Explanation |
|---|---|---|---|
| cultivated land | cultivated land | CU | Farmland for growing crops, such as rice, wheat, and corn, etc. |
| forestland | natural forest | NF | Natural forest growing trees, bamboos, shrubs, not including mangrove. |
|  | plantation | PL | The main crops are palm, coconut, rubber, mango, banana, etc. |
| grassland | grassland | GR | Vegetation with an herbaceous cover of more than 5%. |
| construction land | construction land | CO | Settlement, dock, airport, etc. |
| industrial land | industrial land | IN | Mining, quarrying, and other ground quarry production sites. |

**Table 3.** *Cont.*

| Level 1 | Level 2 | Abbreviation | Explanation |
|---|---|---|---|
| | river | RI | River. |
| water | lake | LA | Lake. |
| | reservoir | RE | Artificially constructed water storage area below the annual water level |
| | mangrove | MA | Evergreen shrubs and small tree communities growing in tidal flats. |
| coastal wetland | salt field | SA | Sites for salt extraction by evaporation. |
| | culture pond | CP | Mostly fishponds. |
| | tidal flat | TF | Intertidal zone between high and low tides. |
| | sea waters | SW | Sea waters. |
| unutilized land | bare land | BA | Bare land that has not been developed or is difficult to develop. |

### 2.3.3. Image Classification Method

For the image classification, we used the following strategies (Figure 3):

First, we used an object-based technique to classify the image. All images were segmented twice before classification. Multi-scale segmentation was performed first. At this stage, the image plaque (level 1) was still relatively fragmented. We used spectral difference segmentation based on the level 1 image objects. Then, a new image object (level 2) was formed, in which some land use types with large patches and homogeneous spectral values, such as mangrove, river, and sea waters were automatically merged.

Second, maximum likelihood supervised classification is known as one of the most effective methods for land use classification. For each land use type, several training samples were selected in eCognition software. Then, by calculating the best separation distance, we chose 6 features for classification: the mean value of the red band/thermal infrared band/mid-infrared band, border index, asymmetry, and main direction. Based on the classification result, we checked and corrected any apparent classification errors manually.

Finally, to validate the precision, approximately 4600 randomly selected reference pixels were used, which meant 100–200 random points for each bay according to the area of bay. The values of the reference point based on the land use classification map were compared to the values based on the Google Earth imagery. The final overall accuracy for each bay was higher than 85%.

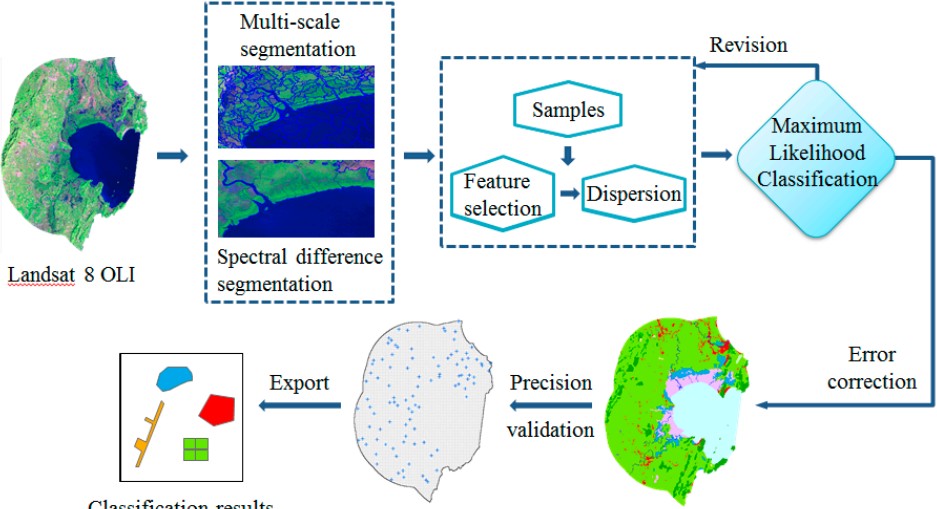

**Figure 3.** Classification flow chart.

#### 2.3.4. Data Analysis

We selected four indexes, including construction index, agricultural land index, forest index, and mangrove index to describe the land use pattern.

(1)    Construction index (CI)

CI = Area of construction land/Area of bay land

We used the ratio of the construction land area and the total bay area to describe CI. CI indicates the regional economic development level. It is closely related to the population density and urban residential point density, etc.

(2)    Agricultural land index (NI)

NI = Area of agricultural land/Area of bay land

The agricultural land includes the culture ponds, cultivated land, and plantations. NI indicates the agricultural economic development level.

(3)    Forest index (FI)

FI = Area of natural forest/Area of bay land

The FI index indicates the percentage of natural forest cover.

(4)    Mangrove index (MI)

MI = Area of mangrove/Area of bay land

The MI index indicates the percentage of mangrove cover.

## 3. Results

### 3.1. Total Trends

Across the region, the total CI in 2018 was 6.6%, which was more than twice the 2.8% CI value in 1988 (Table 4). This was a very large increase which meant that there was a rapid economic development trend. The construction area expanded from 1780 square kilometers in 1988 to 4280 square kilometers in 2018. The net increase in area summed up to 2460 square kilometers, with a high rate of annual increase of 80 square kilometers.

The total NI was 29.8% in 1988; while in 2018, the NI climbed by more than a third, with increases of up to 40.6%. This was a relatively high area of property, which meant that agriculture was flourishing around bays of the South China Sea. The increase in agricultural land area was 7209 square kilometers, which amounted to nearly 240 square kilometers per year. In the composition of agricultural land in 2018, the plantations accounted for the largest proportion (61.9% of the agricultural land), which was followed by cultivated land (24.6% of agricultural land). The plantation areas covered 16.3% of the total land cover in 1988 and the proportion climbed to 25.3% in 2018. The increase in plantation cover contributed 82% of the total increase in agricultural land area. In other words, the growth in agricultural land cover was driven by the growth in plantation cover. The proportion of cultivated land increased slightly from 9.9% to 10.1%. Furthermore, although the area of the culture ponds accounted for a rather small proportion (3.5% in 1988), it increased significantly by 70%. The net increased area summed up to 1145 square kilometers.

The forest resources were very rich in the bays of the South China Sea. In 1988, the natural forest areas covered 37.9% of the total land area around the bays. In 2018, the FI dropped to 25.7%. The natural forest area decreased from 24,390 square kilometers to 16,688 square kilometers. The MI decreased from 15% to 12.7% during the study period. There was a decrease in mangrove cover of approximately 1575 square kilometers, with approximately 53 square kilometers per year.

**Table 4.** Proportion of land use types of each country.

| | CI1988 | NI1988 | | | FI1988 | MI1988 | CI2018 | NI2018 | | | FI2018 | MI2018 |
|---|---|---|---|---|---|---|---|---|---|---|---|---|
| | | CP | PL | CU | | | | CP | PL | CU | | |
| Bay | 2.8% | 3.5% | 16.3% | 9.9% | 37.9% | 15.0% | 6.6% | 5.3% | 25.3% | 10.1% | 25.7% | 12.7% |

### 3.2. Country-Level Trends

The country-level change trends of the four indexes are shown in Figures 4 and 5. The rapid pace of the construction sprawl in the bays occurred in Vietnam, Thailand, the Philippines, and Brunei. The CI in these countries reached approximately 10%; among them, the CI in Thailand nearly doubled. Meanwhile, the NI in Thailand was also the highest among the studied countries. A total of 72.9% of bay land was used for agriculture. The proportion decreased slightly in 2018. The results suggested that the land development intensity in the bays of Thailand was very high, leaving very little land for other uses. In contrast, the NI of the bays in Cambodia and Brunei were low, approximately 10%. The land resources in these areas were relatively abundant.

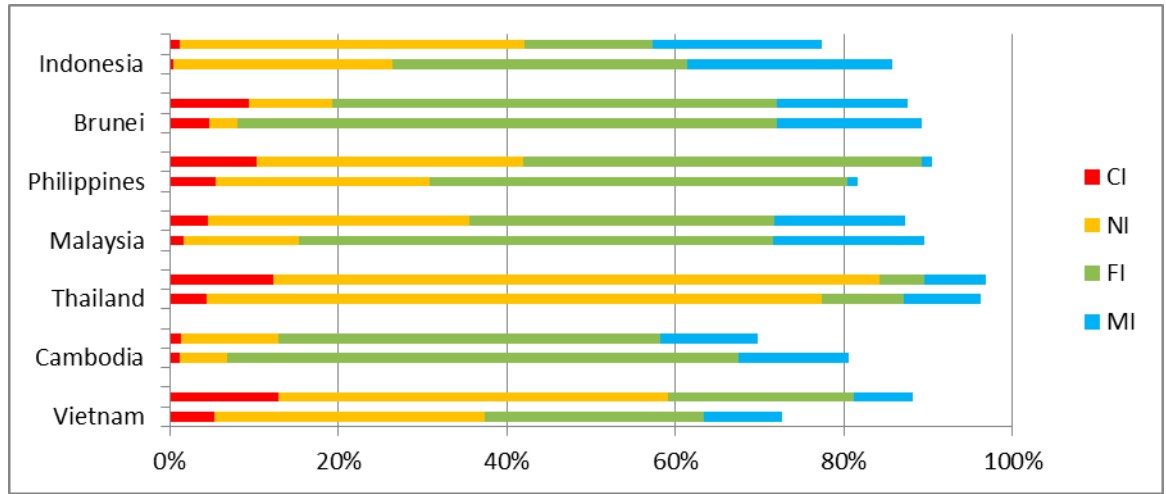

**Figure 4.** Proportion of the land use types in each country in 1988 and 2018. Note: For each country, the upper column represents the indexes in 2018 while the lower column represents the indexes in 1988.

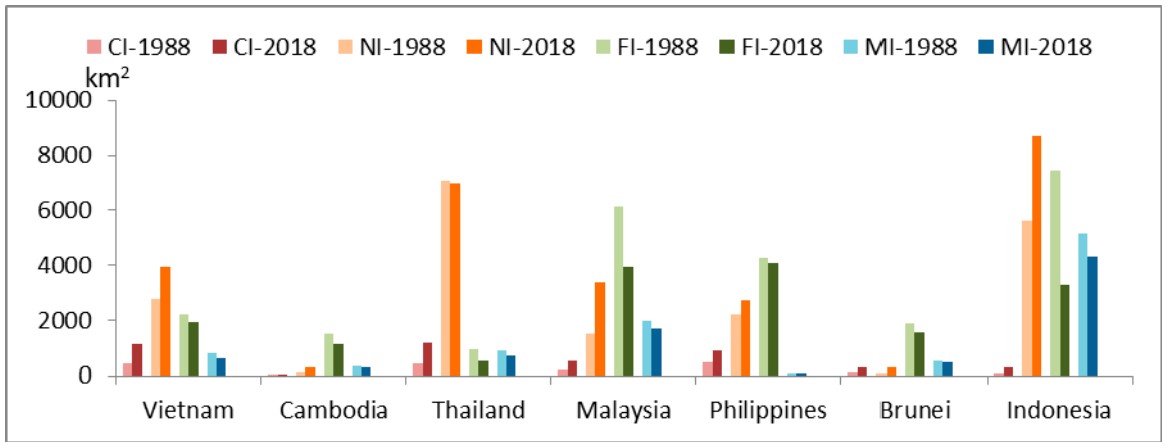

**Figure 5.** Area of the land use types in each country.

The FI in all countries except Thailand were high in 1988. In Cambodia, Malaysia, the Philippines, and Brunei, the FIs were close to or higher than 50%. The percentage of natural forests decreased

severely during the period, especially in Malaysia and Indonesia. The forest area decreased from 6129 square kilometers to 3939 square kilometers and from 7437 square kilometers to 3296 square kilometers, respectively. Mangrove was widespread in bays of Indonesia and Malaysia. The mangrove area in Indonesia decreased from 5165 square kilometers to 4327 square kilometers. The rate of reduction was approximately 28 square kilometers per year, which was more than 50% of the total mangrove loss. The bays in Malaysia, Vietnam, and Thailand followed, and deforestation in these areas contributed 40% of the total mangrove loss.

### 3.3. Bay-Level Trends

The bay-level CIs, NIs, FIs, and MIs were extremely different among the study areas (Figures 6–9). In 1988, a total of 6 bays had CIs of over 5%. Three of these bays were distributed in Vietnam, one bay was in Thailand, one bay was in Malaysia, and two bays were in the Philippines. The highest CI appeared in Hai Loc Bay (16.8%), Dam Ha Trung Bay (12.2%), Danang Bay (11.4%), Bangkok Bay (6.5%), and Manila Bay (17.5%). In 2018, the number of bays for which the CI reached more than 5% increased to 21. The CI of the 5 bays reached more than 20%. These bays were Hai Loc Bay (24.3%), Dam Ha Trung Bay (20.2%), Danang Bay (26.5%), Bangkok Bay (24.0%), and Manila Bay (28.0%). The order of the CI distribution among these bays did not change during the study period. In terms of the growth rates, the CI of Bangkok Bay increased by nearly four times from 6.5% to 24.0%. Second, the CI of Danang Bay doubled from 11.4% to 26.5%. In terms of the spatial distribution, the CIs of the bays in Vietnam were generally high. Except for Rạch Giá Bay, the CIs of the remaining 8 bays exceeded 5%. Among these bays, the CIs of 5 bays exceeded 10%, accounting for five eighths of all bays. In most of the bays in Indonesia and the Philippines, the CIs were less than 2%. These vast regions had low levels of urbanization.

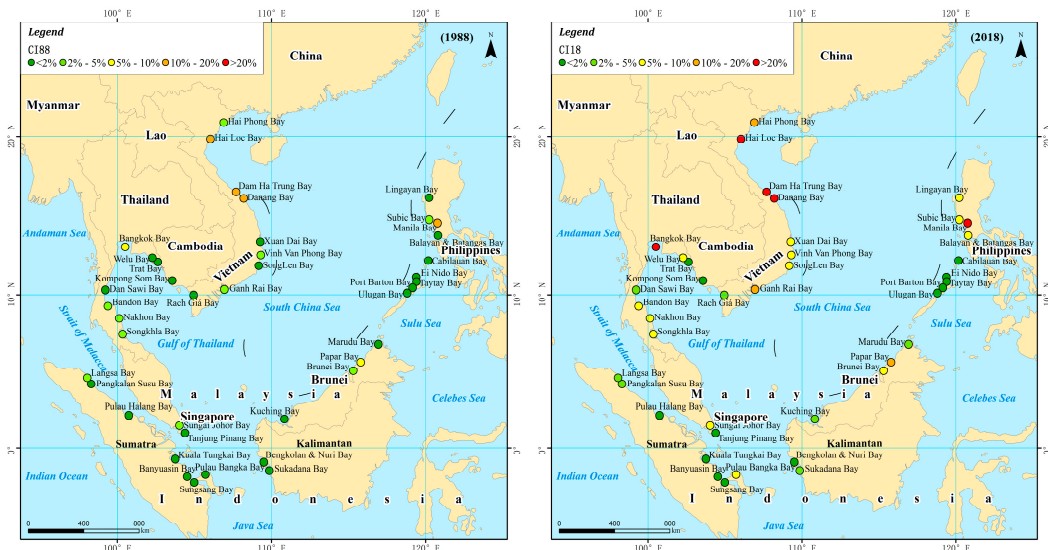

**Figure 6.** CI of the bays in 1988 and 2018.

The distribution of the NIs represented the regional characteristics across the whole study area. The NIs of the bays on the western coast of the South China Sea were significantly higher than those on the eastern coast. Among these areas, the NIs of the bays in the Gulf of Thailand were especially high in both periods, maintaining high proportions of more than 50%. The NIs of Rạch Giá Bay (Vietnam), Dan Sawi Bay (Thailand), and Songkhla Bay (Thailand) reached more than 80%. In addition, the NIs of the bays in northern Sumatra were also high in 2018, over 60%. In Vietnam, the NIs of most bays were distributed from 10% to 50%. In Hai Loc Bay and Rạch Giá Bay, the NIs reached more than 50%. In Kalimantan, the NIs of most bays were distributed from 20% to 40%, which was similar to those

in Vietnam. In the Philippines, the NIs of most bays were less than 10%. Only in the bays of Luzon Island, Philippines, such as Balayan and Batangas Bay, Manila Bay, and Lingayan Bay, did the NIs reach more than 40%. The growth rate was not high, and the NIs decreased slightly in some bays.

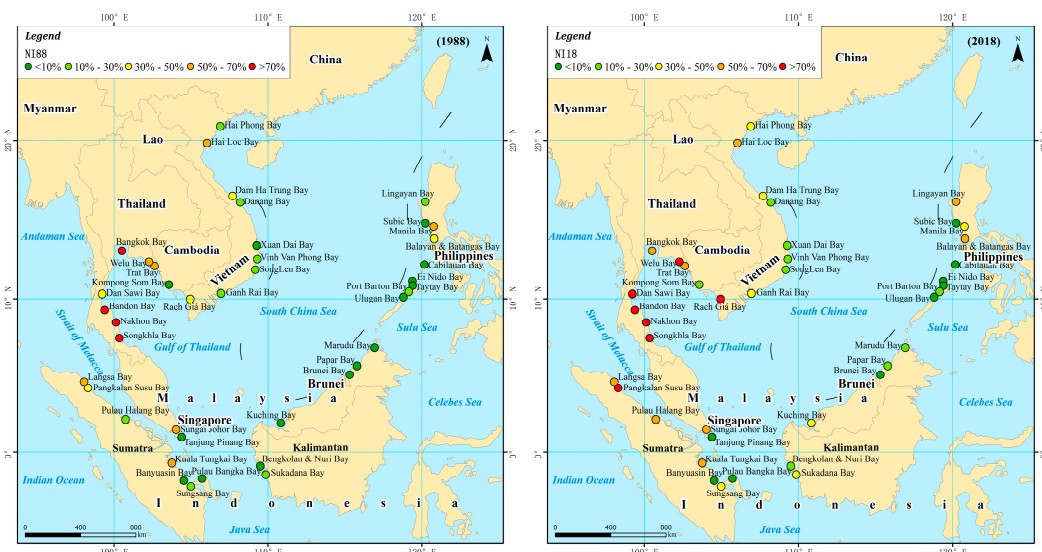

**Figure 7.** NI of the bays in 1988 and 2018.

In terms of the agricultural land composition, the types of agricultural land differed regionally. The main types of agricultural land are shown in Figure 8. The sizes of the circles represent the ratio of agricultural land area to total bay area. Plantation land and cultivated land were the two main agricultural land types within both periods. Bangkok Bay was the only bay dominated by culture ponds in 2018. In terms of the spatial distribution, the bays with cultivated land as the dominant agricultural land type were concentrated in the northern South China Sea (Vietnam and Philippines). The plantation-dominated bays were mostly distributed in the southern South China Sea (Gulf of Thailand, Malay Peninsula, Sumatra, and Kalimantan). The conversion of the agricultural land over the period was not obvious, mainly in the Gulf of Thailand. In Nakhon Bay and Songkhla Bay, the dominant agricultural land type changed from cultivated land to plantation land.

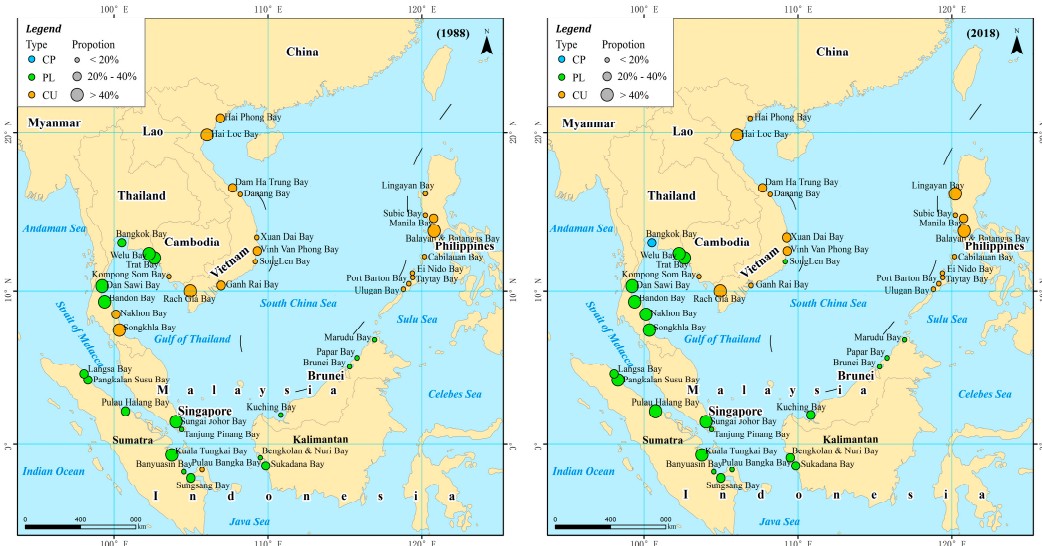

**Figure 8.** Type and percentage of agricultural land in 1988 and 2018.

The spatial distribution of the natural forest proportion showed concentrations in Kalimantan in Indonesia and Palawan in the Philippines (Figure 9). The FIs were generally higher than 50%. In addition, the FIs were relatively high in the bays of central Vietnam. Over the 30-year study periods, the forest cover losses mostly occurred in bays of Sumatra and western Kalimantan. In the bays with FIs of over 50%, including Tanjung Pinang Bay, Bengkolan and Nuri Bay, Sukadana Bay, and Kuching Bay, the forest cover losses exceeded 50%. In Pangkalan Susu Bay and Pulau Halang Bay in western Sumatra, the forest percentage decreased to lower than 10%.

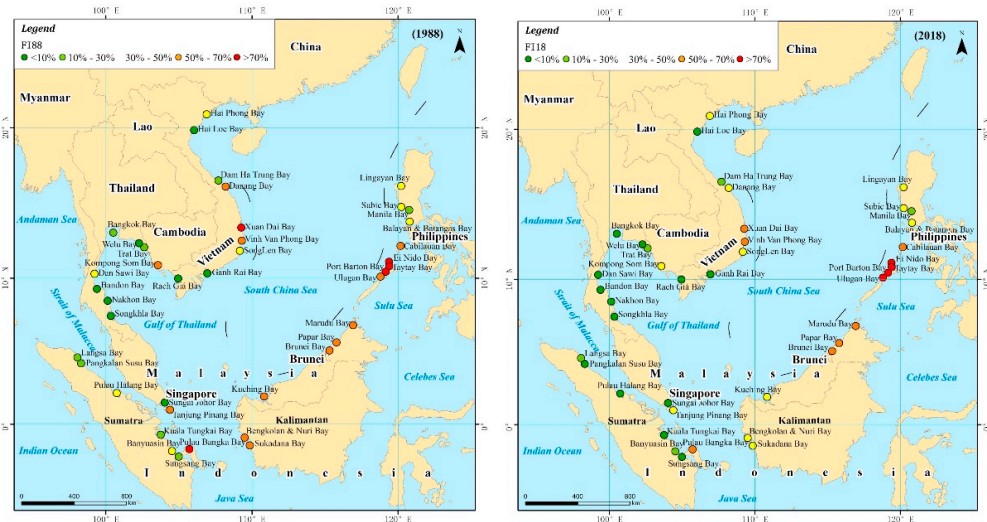

**Figure 9.** FI of the bays in 1988 and 2018.

The distribution of the MIs also showed regional differences across the whole study area (Figure 10). The MIs of most bays in Sumatra and Kalimantan were high, from 15% to 30%. The decline rates of MI in most bays were lower than 20%. There were some exceptions. The MI of Pangkalan Susu Bay (Sumatra) dropped from 31.3% to 15.8%. The MI of Hai Phong Bay (Vietnam) decreased from 11.1% to 4.7%. Meanwhile, the mangroves also spread throughout most bays in the Gulf of Thailand and the Malay Peninsula. The MIs of these bays ranged from 5% to 15%. The MIs of the bays in Vietnam and the Philippines were generally low, except for Ganh Rai Bay in Vietnam.

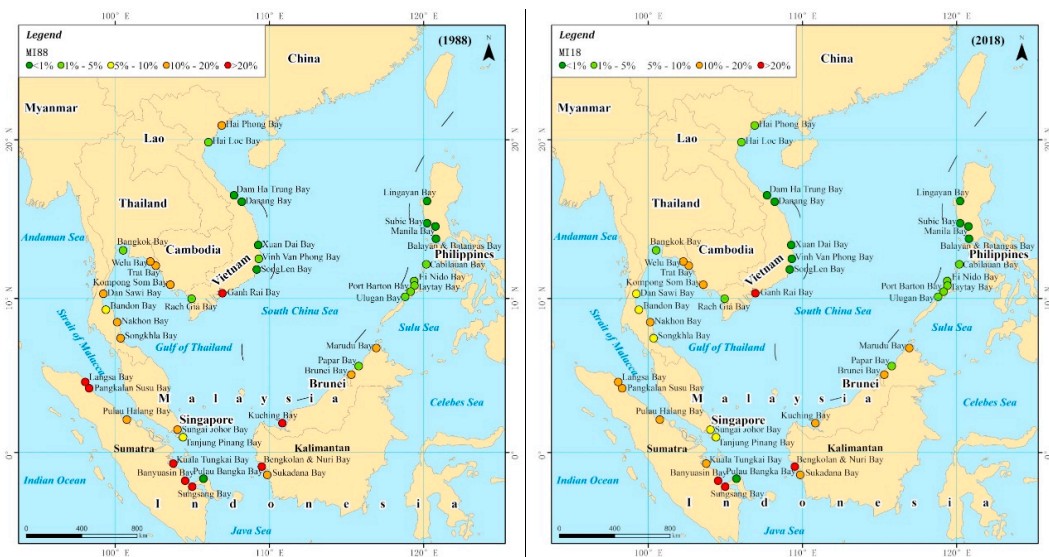

**Figure 10.** MI of the bays in 1988 and 2018.

On the bay-level, the distribution of land use conversion differed extremely (Figure 11). Land use conversion mostly occurred in the bays on the western coast of the South China Sea. The bays with land use conversion proportions in excess of 50% were distributed in Ganh Rai Bay and Rach Giá Bay in Vietnam, as well as Pangkalan Susu Bay and Pulau Halang Bay in Sumatra. Furthermore, the spatial distribution of the types of land use change presented regional characteristics. The replacement of natural forest by plantation was the dominant type of land use change throughout the region, concentrated in Sumatra and Kalimantan. The conversion of cultivated land to plantation land was identified mostly in the Gulf of Thailand. The conversion patterns of cultivated land to construction land and natural forest to cultivated land, appeared in Vietnam and the Philippines.

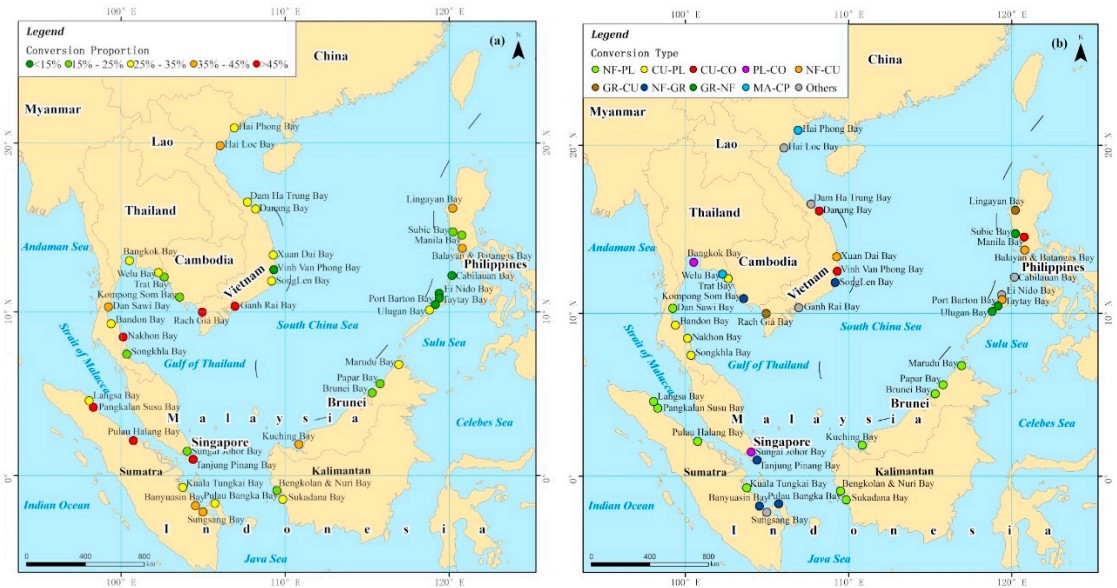

**Figure 11.** Overview of the land use conversion distribution. (**a**) The proportion of converted land area, (**b**) dominant land use conversion type.

## 4. Discussion

### 4.1. Expansion of Plantation

In the bays surrounding the Southeast Asia, plantation land was the second most common land use type after natural forest and the expansion of plantations was notable. The total area of plantation land had increased by 57% in 30 years. The rapid expansion of plantations was due to the strong demand from the world economic crop market. Over the past half century, Southeast Asia has witnessed major shifts from predominantly subsistence agrarian economies to increasingly commercialized agriculture [41]. The production pattern of export-oriented crops (such as oil palm, rubber, coffee, pepper, cocoa, and farmed shrimp) have led to extraordinary booms in the numbers of large estates as well as in those of smallholders [42,43]. Plantation-dominated bays were mostly distributed in the southern South China Sea (Gulf of Thailand, Malay Peninsula, Sumatra, and Kalimantan). It is determined by the climate conditions in Southeast Asia. The tropical monsoon climate with constant heavy rains in these areas contributed a lot to the growth of economic plantations [44]. Also, it is influenced by the local policies. Natural rubber is an important industrial raw material which is made from rubber trees. Approximately 35% of the latex production in the world is from Thailand [45]. Palm oil is the most important vegetable oil in the world. Indonesia has been responsible for the majority, supplying roughly 60% of the total annual increase in palm oil production. With current annual rates of increase, Indonesia could theoretically double current national oil palm acreage by 2030 [46].

Planted forest area in Southern and Southeast Asia increased by 85% from 1990 to 2015 [16]. In contrast, the growth rate of plantation land in the bays surrounding the South China Sea was

lower. But, the proportion of plantation land in the bays was much higher and plantation expansion was the most significant land use change phenomenon across the bays. The top 10 land use change types are shown in Figure 12 by area. New plantation lands were converted from natural forest, grass land, cultivated land and mangrove (ranking 1st, 4th, 5th, and 6th in Figure 12). Among them, natural forests contributed more than 60% of the increased plantation area. Among the forest types, the studies showed that peat swamp forest loss was the most serious and was closely related to the oil palm expansion [17,27]. The traditional planting methods of using fire to prepare lands for plantation agriculture brought about lots of environment problems, such as huge amounts of carbon emissions [47]. Plantation expansion in Kalimantan is projected to contribute 18–22% of the Indonesia's 2020 CO2-equivalent emissions [48]. In addition, the conversion from cultivated land to plantation was significant in the Gulf of Thailand. Most traditional rubber plantation areas are in the southern regions of Thailand [35]. Thailand is one of the main rice-producing countries and the rice harvested areas are located in the northeastern regions. The proportion of rice harvested areas in the southern areas is lower than 5% and decreasing year by year [49]. The Thailand government has propped up the rubber industry through a series of policies, including a minimum tax on the rubber industry [50]. In Phatthalung Province of the southern Thailand, approximately 25% of the lowland rice fields were converted to rubber plantations between 1976 and 1990, and the conversion rate reached 50% between 1990 and 2006 [51].

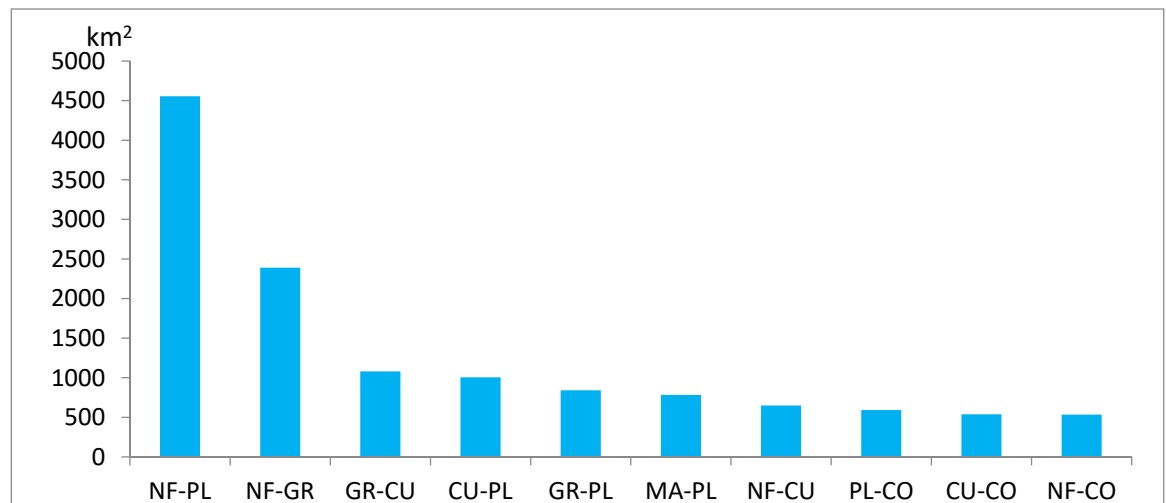

**Figure 12.** Area of the land use change types.

*4.2. Deforestation*

Deforestation is a major concern globally. The forest loss in Southeast Asia accounted for 21% of the total forest loss in the world from 1990 to 2015. The annual loss rate was 0.34% [52]. The forest loss was attributed to commodity driven deforestation, forestry, shifting agriculture, and wildfires [53].

In the study area, the natural forest area decreased by 31.6% from 37.9% in 1988 to 25.7% in 2018. The net forest cover loss was 7702 square kilometers in 30 years. In terms of spatial distribution, the natural forest loss was mostly identified in the bays of central Sumatra Island and western Kalimantan Island. Related studies also confirmed that the eastern lowlands of Sumatra and the peatlands of Sarawak, Borneo were the main areas of forest loss [27]. Commodity driven deforestation for plantations was the major driver, accounting for 53% of the natural forest loss (Table 5). Together, 4555 square kilometers of natural forest were cut down and replaced with commercial forests, such as planted oil palms, rubber, and coconuts. Sumatra has always been a key area for oil palm production in Indonesia, with Riau as the leading province [54]. The forest cover loss in Riau contributed half of the total forest loss in Sumatra [55]. Second, logging and wildfire were another major cause, accounting for 31% of the forest loss. Large amounts of the natural forest were converted to grass land or bare land,

perhaps as land reserved for plantations or abandoned land. In Katapang, Indonesia, fire was the cause of 90% of deforestation between 1989 and 2008, and 20% of wildfires across Indonesia can be attributed directly to oil palm plantation practices [56]. Furthermore, extensive logging contributes a lot to carbon emissions and may increase wildfire risks [17]. The wildfire will cause new forest losses. In addition, approximately 15% of the natural forest land was exploited for cultivated land and construction.

The annual natural forest loss rate of the bays surrounding the South China Sea was 1.05%, which was much higher than the annual loss rate in Southeast Asia. This finding is worthy of considerable attention. Forest loss is a serious threat to biodiversity and agricultural productivity and contributes to global warming [57–59]. In the process of the rapid economic development of bays, decision makers should pay more attention to forest protection.

**Table 5.** Natural forest cover and change from 1988 to 2018.

|  | Area ($km^2$) |
| --- | --- |
| Replaced by plantation | 4555.3 |
| Conversion to grass/bare land | 2615.6 |
| Conversion to cultivated land | 650.0 |
| Conversion to construction | 534.9 |
| Conversion to other types | 190.8 |
| Afforestation or forest restoration | 844.3 |
| Gross forest cover loss | 8546.5 |
| Gross forest cover gain | 844.3 |
| Net forest cover loss | 7702.3 |

### 4.3. Mangrove Deforestation

The problem of mangrove degradation is worthy of attention. Mangroves are characteristic forest communities on tropical and subtropical coasts and have important ecological significance. It is well known that Southeast Asia is one of the most important mangrove habitats in the world, covering an area of 63,795 square kilometers in 2013 (FAO, 2019). Now, mangrove forests are threatened by factors including the expansion of shrimp farming, oil palms, overexploitation, and sea level rise [28,60–62].

Richards et al. [28] analyzed the rates and drivers of mangrove deforestation in Southeast Asia between 2000 and 2012. Research has shown that mangroves decreased by 2.12% (an average of 0.18% per year) and that aquaculture is a major pressure on mangrove systems. In this study, the mangrove deforestation in the bays surrounding the South China Sea was more serious. The percentage of mangrove cover decreased from 15% to 12.7% during the study period. The magnitude of the net loss was 16.2%, covering an area of 1575 square kilometers. The magnitude of the average annual loss was 0.54%, tripling the total mangrove deforestation in Southeast Asia. Meanwhile, the decrease in mangroves was partly offset by the expansion of mangroves (Table 6). It was notable that nearly half of mangrove recovery or regrowth occurred in tidal flats. Mangrove expansion in estuaries and offshore waters nearly contributed to the remaining half of the growth.

An analysis of the mangrove change patterns was performed. The expansion of plantations was suggested as the major cause of mangrove loss, accounting for 41% during the study period. Massive plantation cultivation had severe impacts on the mangrove ecosystems, especially in Malaysia and Indonesia. Second, aquaculture (culture pond) was another major reason, accounting for 21% of the mangrove loss. The South China Sea is rich in fish resources. The 2010 FAO statistical data showed that the catches in Indonesia, Vietnam, the Philippines and Malaysia ranked 3rd, 10th, 11th, and 16th respectively, in the world [63]. The loss of mangroves was related to the prevalence of offshore fish farming, especially in estuaries such as Ganh Rai Bay (Vietnam), Welu Bay (Thailand), and Pangkalan Susu Bay (Indonesia). Finally, urbanization and cultivation together contributed to 11% of the mangrove loss.

**Table 6.** Mangrove cover and change from 1988 to 2018.

|  | Area (km$^2$) |
|---|---|
| Replaced by plantation | 782.5 |
| Conversion to culture pond | 401.8 |
| Conversion to cultivated land | 97.8 |
| Conversion to construction | 110.2 |
| Conversion to other types | 525.4 |
| mangrove restoration | 467.8 |
| Gross mangrove loss | 1917.7 |
| Gross mangrove gain | 467.8 |
| Net mangrove loss | 1449.8 |

## 5. Conclusions

The objective of this study was to provide a uniform assessment of the LUCC in the large bays surrounding the South China Sea. Our study showed that dramatic changes took place from 1988 to 2018.

First, the total construction land increased more than double and the rapid sprawls was concentrated in Vietnam, the Gulf of Thailand, and Luzon Island in the Philippines. Second, the agricultural land cover increased from 29.8% to 40.9% and the expansion of plantations played the most important role. Plantation expansion was the most significant phenomenon of land use change across the bays, concentrated in the Gulf of Thailand, Sumatra, and western Kalimantan. Third, as the main land use type, the natural forest loss was serious, with a high pace of 31.6%. Serious deforestation was identified in Sumatra and western Kalimantan. Commodity-driven deforestation for plantations was the major reason, accounting for 53% loss of the natural forest. Finally, the percentage of mangrove cover decreased from 15% to 12.7% over the study period. The expansion of plantations' expansion was the major cause of mangrove loss, followed by aquaculture.

The study contributed to a better understanding of the LUCC patterns in the bays surrounding the South China Sea. Agriculture is highly developed in the bays and plantations play an important role in agriculture. The rates of natural forest loss and mangrove loss were serious, compared to those in Southeast Asia. Forest protection is worthy of more attention, especially in the bays. Subject to the image resolution and classification techniques, the classification categories were relatively rough. In future studies, there is the need to focus on: (i) the subdivision of plantations and interference of plantations with natural forests, and (ii) the potential of plantation expansion in the future.

**Author Contributions:** J.Z. and F.S. wrote the paper. All authors have read and agreed to the published version of the manuscript.

**Funding:** This work was funded by the National Natural Science Foundation of China (NSFC) (41890854).

**Conflicts of Interest:** The authors declare no conflict of interest.

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
