# Peer review of "Land Use Change in the Major Bays Along the Coast of the South China Sea in Southeast Asia from 1988 to 2018"

_land, doi:10.3390/land9010030_

Round 1

Reviewer 1 Report

Dear Authors,

I revised your manuscript about the land cover change analysis of South China bays. According to my opinion, the following issues you be addressed (I also reported some comments in the uploaded pdf file to your consideration):  

Abstract.

line 14: before the result highlights, you should introduce the applied methods as well as the principal analysis

Keywords: avoid acronym in keywords

Introduction.

Line 24-27: these three definitions of bay are totally unconnected to the followed text, without references. In my opinion, should be better described the geomorphology of a bay in respect to the type of study of this paper. Therefore, I suggest to avoid generic definitions

Lines 49-70: here and below you've reported several localities related to different studies but , in my opinion, readers could be confused about this. you could add a map of the case studies and a comparison of their trends in a table or a graph. Furthermore, in the present form, you reported a lot of information about different studies without explain in depth the aim of these information

There are lacks of references in general in this section

The end of Introduction should include the aspect of relevance and importance of your research. There was a lot of work done, so it is worth to emphasize why it is important.

Study Areathere is lack of references to presented information.

2.3.2. Land use classification system: Did you adopt a literature methodology? Is this classification already in use? Why did you choose only 15 land classes? please consider to include more information about this part and the methodology in order to give all parameters and reasons that you’ve taken in the construction of land use maps. You can add a useful example here: 1.              Corbau, C.; Zambello, E.; Rodella, I.; Utizi, K.; Nardin, W.; Simeoni, U. Quantifying the impacts of the human activities on the evolution of Po delta territory during the last 120 years. J. Environ. Manage. 2019, 232, 702–712.

line 181-182: I really don't understand the comparison of a  urbanization ratio and GDP, which are calculated with different methods and have different purposes...

Fig 6,7,8,9,10 have low resolution and legends/points, place names are too small

Please don’t confuse results and discussion (see the pdf file). Discussion section is limited to few sentences mixed with other results. Therefore, there is lack of discussion. Please try to refer to works of other researchers, to compare your results with obtained by them.

Conclusion: 1) summarizing what you did in the paper, including its main findings, 2) acknowledging the limitations of your work and 3) proposing steps for future research that builds on what you’ve done in the paper. Moreover, avoid specific results

The follow references may take into consideration: 

- Copernicus
https://land.copernicus.eu/user-corner/technical-library/Nomenclature.pdf (2012)
- Y. Ye, B.A. Bryan, J. Zhang, J.D. Connor, L. Chen, Z. Qin, M. He
Changes in land-use and ecosystem services in the Guangzhou-Foshan Metropolitan Area, China from 1990 to 2010: implications for sustainability under rapid urbanization
Ecol. Indicat., 93 (2018), pp. 930-941, 10.1016/j.ecolind.2018.05.031
- Z. Wu, Z. Yu, X. Song, Y. Li, X. Cao, Y. Yuan
A methodology for assessing and mapping pressure of human activities on coastal region based on stepwise logic decision process and GIS technology
Ocean Coast. Manag., 120 (2016), pp. 80-87, 10.1016/j.ocecoaman.2015.11.016
- V. Parravicini, A. Rovere, P. Vassallo, F. Micheli, M. Montefalcone, C. Morri, C. Paoli, G. Albertelli, M. Fabiano, C.N. Bianchi
Understanding relationships between conflicting human uses and coastal ecosystems status: a geospatial modeling approach
Ecol. Indicat., 19 (2012), pp. 253-263, 10.1016/j.ecolind.2011.07.027

Author Response

We appreciate all the comments from the reviewers on the above manuscript. Revisions have been made and the modifications were marked red in the paper.

Reviewer 2 Report

This study considers trends in land use in large bays in the South China Sea between 1988 and 2018. Landsat data from those two time points is used in the analysis. Since this region is experiencing huge economic growth, and land use change, and bays are particularly vulnerable to direct and secondary impact from land use, this is a useful and timely subject for study.

Overall, I would describe the manuscript and study as rudimentary, but functional. Only the two time points at the beginning and end of the study period were utilized, and the analysis primarily consists of describing the observed changes. The presentation of the results is general. However, I acknowledge that the study covers a large geographical extent, and will likely be of interest as a starting point for further, more specific studies of the land-use changes.

The quality of the writing is variable. I identified many errors in the first 100 lines (detailed below), but it was too time-consuming to identify every mistake throughout the manuscript. Therefore, after the first 100 lines I have only listed mistakes that seriously harmed the understanding of the manuscript. I would advise the authors to carefully check the writing, seeking outside help if necessary.

The validation of the classification seems robust in terms of the sample sizes used. I was left wondering if there were any places it performed differently, whether better or worse, since the manuscript just says that it was >85% in all places. 15% error would be quite a lot in relation to some of the land use changes presented as results, so it did leave me curious as to whether there were any areas the authors were less confident about their results. An uncertainty map could address this, though I would not require it of them.

Here are some minor comments and queries:

Line 26: What does “As pearls in the coastal zone” mean? Is this a metaphor? Or is this a reference to something else? When including citations, it should be the surname only, and not the initial. I have provided a few examples in the corrections below (lines 46 and 68). It needs to be corrected throughout (lines 74 and 77, for example). Throughout the manuscript “et al.” is used to mean its literal translation of “and others”. However, when implying a list of items could be extended it is best to use “etc.”, especially in a scientific publication where “et al.” is usually reserved for implying additional authors in citations. Most of the figure captions need spaces between the “Figure” and the number. What does “Part of additional revision work was done after classification.” mean in line 169? Were the model parameters or variables changes after viewing the initial results? That could create a multiple comparisons problem. Figures 6-9 use a base map. Where is it from? Some place names have clearly been added (with a white stroke around them), others (presumably on the base map) were illegible. “Thiland” should be “Thailand”. In line 404, commodity driven deforestation is described as “natural forest loss”. Is it really natural?

Minor Corrections (thorough up to line 100, only major corrections thereafter):

Line 12: “Asia- pacific” to “Asia-pacific”

Line 14: “patter” to “pattern”

Line 14: “dominated” to “dominant”

Line 15: “2rd” to “2nd”

Line 34: “et al.” to “etc.”?

Line 36: “Study on” to “Study of”

Line 36: “lucc” to “LUCC”

Line 42: Either capitalize “academy” and “sciences”, or remove the capitalization of “National”

Line 44: Either capitalize “lucc” here, or remove the capitalization in Line 38, and elsewhere.

Line 46: “Schneider A et al.” to “Schneider et al.”

Line 47: “Result” to “Results” or “The results”

Line 50: “continues” to “continuous” or “continuing”

Line 51: “increased 4.8 times” to “increased to 4.8 times”

Line 53: “The need of” to “The needs of” or “The need for”

Line 54: “coast area” to “coastal areas”

Line 55: “limited to” to “limited to,”

Line 57: “most host topics” to “most common topics”?

Line 58: “took important place” to “took an important place”

Line 61: “with high growth” to “with a high growth”

Line 68: “Li Z et al.” to “Li et al.”

Line 70: “most of new” to “most of the new”

Line 73: “deforestation rate of the Southeast Asia” to “the deforestation rate of Southeast Asia”

Line 74: “250-mspatial” to “250-m spatial”

Line 76: “peatswamp” to “peat swamp”

Line 80: “Papua New Guinea) and” to “Papua New Guinea and”

Line 81 onwards: “Million” starts being capitalized here. Please correct throughout.

Line 89: “Kuching et al” seems out of place here in the list, and should be “Kuching et al.”

Line 89: “were serious” to “were seriously”

Line 92: “not include” to “not including”

Line 95: “in details” to “in detail”

Line 95: “on region level” to “on a regional level”

Line 97: “Southeast Asia” to “of Southeast Asia”

After the introduction, I no longer identified all corrections.

Line 117: “expected amount” to “expected rainfall”

Line 151: “coastline lied” to “coastline lay”

Line 152: “analysis” to “analyze”

Line 164: “merged .” to “merged.”

Line 165: “Likelihood” should not be the only word capitalized.

Line 197: Consider “significant” to “large” or similar

Line 197: “economic a rapid economic” to “a rapid economic”

Line 203: “was7,209” to “was 7,209”

Line 206: “covers” to “area”

Line 209: “litter” to “small”

Line 210: “area culture” to “area of culture”

Line 214: “As regards” to “In regard to”

Line 221: “increased nearly twice” to “nearly doubled”

Line 237: “Bay- level” to “Bay-level”

Line 271: “Tik.Ara bay” to “Tlk. Ara bay”

Line 287: “Tik.Ara bay” to “Tlk. Ara bay”

Line 314: “Terrain” to “terrain”

Line 326: “crop” to “crops”

Line 351: “Peat” to “peat”

Line 358: “crease” to “increase”

Line 387: “Tik.Ara bay” to “Tlk. Ara bay”

Line 388: “in together” to “together”

Line 397: “increased more than twice” to “more than doubled”

Line 404: “major drive” to “major reason”?

Line 408: “plantation” to “plantations”

Line 410: “It was worth of more attention to forest protection” to “Forest protection is worthy of more attention”

Author Response

(The authors gave the same response as above.)

Round 2

Reviewer 1 Report

Dear authors, 

I've read the second version of your paper and its improvements. 

I also suggest to rewise: 

fig 4 and 5 adding the acronym decriptions (i.e. CI, NI, FI, MI) legends and text of figg. 6, 7, 8, 9, 10, 11 are always too small for a correct reading. Please try to resize these figures. 

Good luck! 

Author Response

(The authors gave the same response as above.)

Reviewer 2 Report

Thank you to the authors for their careful attention to my suggestions and corrections, and their thorough responses to my comments. I am satisfied with the authors' responses to all of the issues over which I raised questions in my original review.

I went through the sections added in the modified version of the manuscript and identified a number of corrections, and some systematic mistakes that need careful attention, as follows:

There appear to be several instances of missing spaces before reference numbers, for example line 27, line 28, and line 34.

Line 12: “Asia-pacific” to “Asia-Pacific”?

Line 25: “The bay is a water area deep into the mainland and islands.” Is this meant to mean that the bay extends into the mainland and islands? Or surrounds them? Or encompasses them? Some rephrasing is needed, but I was not sure of the intended meaning.

Line 27: “etc[1].” to “etc. [1].” Check for more instances of missing abbreviations of etc.

Line 28: “land[2]” to “land [2]” Check for more instances of missing spaces.

Line 60: “area” to “areas” or “coastal area” to “the coastal area”

Line 85: “Stibig[24]” to “Stibig et al. [24]”

Line 96: “results of literature” to “results of a literature”

Line 171: “there’re” to “there are”

Line 172: “etc[39,40].” to “etc. [39,40].”

Line 188: “Then new image” to “Then a new image”

Line 189: “value” to “values”

Line 268: “In 1988, together 6 bays” to “In 1988, a total of 6 bays”

Line 340: “2nd land use type” to “2nd most common land use type” or “2nd largest land use type”

Line 341: Consider “notable” instead of “amazing”.

Line 347: “in southern of” to “in southern regions of”

Line 356-357: “By contrast” to “In contrast”

Line 367-368: Sentence starting “Besides, the conversion” needs rephrasing. I am not sure of the intended meaning, but perhaps: “The conversion of cultivated land to plantation was significant in the Gulf of Thailand.”?

Line 370: “in Northeastern” to “in the north-east” or “in the Northeastern regions”

Line 371-372: “the rubber industry” repeated by mistake.

Line 384: “In spatial distribution” to “In terms of spatial distribution”

Line 391: “contributed to half” to “contributed half”

Author Response

(The authors gave the same response as above.)
